# Regular Training Increases sTWEAK and Its Decoy Receptor sCD163–Does Training Trigger the sTWEAK/sCD163-Axis to Induce an Anti-Inflammatory Effect?

**DOI:** 10.3390/jcm9061899

**Published:** 2020-06-17

**Authors:** Robert Schönbauer, Michael Lichtenauer, Vera Paar, Michael Emich, Monika Fritzer-Szekeres, Christoph Schukro, Jeanette Strametz-Juranek, Michael Sponder

**Affiliations:** 1Department of Cardiology, Medical University of Vienna, Währinger Gürtel 18-20, 1090 Vienna, Austria; robert.schoenbauer@meduniwien.ac.at (R.S.); christoph.schukro@meduniwien.ac.at (C.S.); 2Department of Cardiology, Clinic of Internal Medicine II, Paracelsus Medical University of Salzburg, 5020 Salzburg, Austria; m.lichtenauer@salk.at (M.L.); v.paar@salk.at (V.P.); 3Austrian Federal Ministry of Defence, Austrian Armed Forces, 1090 Vienna, Austria; dr@emich.at; 4Department of Medical-Chemical Laboratory Analysis, Medical University of Vienna, 1090 Vienna, Austria; monika.fritzer-szekeres@meduniwien.ac.at; 5Rehabilitation Centre Bad Tatzmannsdorf, 7431 Bad Tatzmannsdorf, Austria; jeanette.strametz-juranek@pensionsversicherung.at

**Keywords:** TWEAK, CD163, inflammation, atherosclerosis, physical activity, sports

## Abstract

Background: Low levels of soluble tumor necrosis factor-like weak inducer of apoptosis (sTWEAK) were reported in patients with coronary artery disease, heart failure, chronic kidney disease and diabetes mellitus. Soluble cluster differentiation 163 (sCD163) serum levels are related to M2 macrophages, having anti-inflammatory attributes. As sport is well-known for its anti-inflammatory and cardioprotective effects we aimed to investigate the influence of eight months of physical activity on serum sCD163 and sTWEAK levels. Methods: In total, 109 subjects with at least one cardiovascular risk factor were asked to perform endurance training within the calculated training pulse for eight months. Overall, 98 finished the study. The performance gain was measured/quantified by bicycle stress tests at the beginning and end of the observation period. The cohort was divided into four groups, dependent on their baseline performance and performance gain. sCD163 and sTWEAK were measured at baseline and after two, six and eight months by ELISA. Results: Those participants who had a performance gain of ≤2.9% (mean gain 12%) within eight months showed a significant increase in sTWEAK (group 2: from 133 to 200 pg/mL, *p* = 0.002 and group 4: from 166 to 212 pg/mL, *p* = 0.031) and sCD163 levels (group 2: from 255 to 348 ng/mL, *p* = 0.035 and group 4: from 247 to 288 ng/mL, *p* = 0.025) in contrast to subjects without performance gain (sTWEAK: group 1: from 161 to 177 pg/mL, *p* = 0.953 and group 3: from 153 to 176 pg/mL, *p* = 0.744; sCD163: group 1: from 289 to 256 ng/mL, *p* = 0.374 and group 4: from 291 to 271 ng/mL, *p* = 0.913). Baseline sCD163 correlated with erythrocyte count, hematocrit, ASAT and lipoprotein a, the presence of hypertension and a BMI > 30 kg/m^2^. Conclusion: Regular physical activity leads to a significant increase in sCD163 and sTWEAK levels of up to 37% and 50%, respectively. It is well-known that physical activity prevents or retards the onset and genesis of chronic inflammatory disease. One possible way of how training evolves its beneficial effect might be by modifying the inflammation status using the sTWEAK–sCD163 axis. Brief Summary: Regular physical activity leads to a significant increase in sTWEAK and sCD163 levels. Both factors are diminished in patients with chronic (inflammation-based) diseases, such as coronary artery disease, heart failure, pulmonary artery hypertension, chronic kidney disease and diabetes mellitus. It seems that the amounts of soluble TWEAK and CD163 are essential for a healthy balance and modulation between pro- and anti-inflammatory processes, and regular physical training could use the sCD163–sTWEAK axis to unfold its beneficial effect.

## 1. Introduction

By attenuating the negative effects of cardiovascular risk factors, regular physical activity counteracts or retards the onset and progression of atherosclerosis, respectively [1]. Chronic inflammation is considered the basic pathomechanism of atherosclerotic diseases and a vast number of biomarkers and pathways were proposed to be involved in the process of atherosclerosis, such as tumor necrosis factor-like weak inducer of apoptosis (TWEAK) and cluster differentiation 163 (CD163) and their soluble forms, sTWEAK and sCD163. TWEAK is a secreted pro-inflammatory multifunctional ligand of the tumor necrosis factor super family and was first described in 1997 [2]. The binding of TWEAK to its initial receptor, Fn14 (fibroblast growth factor inducible molecule 14), which is mostly expressed in injured tissue, triggers an inflammatory response (e.g., by inducing interleukin 6 and 8 (IL-6/8) [3]) which has been shown to be involved in the formation and progress of several chronic inflammatory diseases, such as atherosclerosis [4]. However, CD163 is a further receptor of TWEAK but a scavenger receptor (meaning that the binding of TWEAK to CD163 leads to the neutralization of both molecules) and an important modulatory cytokine of inflammation [5]. In vitro, it was shown that CD163-expressing macrophages reduce endogenous TWEAK expression and consequently induce the degradation of TWEAK [6]. This reaction might lead to attenuated inflammation in atherosclerosis [7]. 

After the elimination of classic cardiovascular risk factors, physical activity might be the most potent “remedy” against a chronically increased inflammatory status as it occurs in atherosclerosis. sTWEAK and sCD163 might be involved in this process and, therefore, it was the aim of the present prospective observational study to investigate the influence of eight months of training on sTWEAK and sCD163 levels in serum.

## 2. Participants, Material and Methods:

### 2.1. Study Population

A total of 109 subjects were recruited with the following inclusion criteria: age 30–65 years, the physical ability to perform a bicycle stress test and endurance training and the presence of at least one classic cardiovascular risk factor defined as follows: overweight (body mass index (BMI) > 25.0 kg·m^2^), hypertension (systolic blood pressure (SBP) > 140 +/− diastolic blood pressure (DBP) > 85 mmHg at rest/antihypertensive medication), hyper/dyslipidemia (anamnestic therapy with statins), diabetes mellitus (HbA1c > 6.5 rel%/antidiabetic medication), current smoking, known chronic heart disease (CHD) (anamnestic myocardial infarction (MI), percutaneous coronary intervention (PCI), coronary artery bypass graft (CABG) or stroke) and a positive family anamnesis for MI/CVD/stroke of the mother and/or father. Exclusion criteria were: age < 30 or > 65 years, no ability to perform a bicycle stress test and endurance exercise, current infectious and/or oncologic disease (anamnestic or increased baseline infection lab parameters). The weekly alcohol intake was measured/quantified by units: 1 unit corresponds to 0.33 mL beer, 0.125 mL red/white wine or 0.02 mL spirits. 

Anamnesis and a physical examination, including the measurement of anthropometric data (height, weight, body water, muscle mass or fat with a diagnostic scale, Beurer BG 16, Beurer GmbH, Ulm, Germany), were done at the first study meeting. All of the participants were tested by a bicycle stress test to define their initial performance level. All of the subjects were asked to perform physical activity for at least 150 min/week at moderate and/or 75 min/week at vigorous intensity according to their individual training pulse (using the Karvonen formula with an intensity level of 65–75% for moderate and 76–93% for vigorous intensity). The participants were free to choose the form of sport. Strength training was not mandatory. To verify and quantify the effectivity of their training progress, over the observation period a second bicycle stress test was performed after eight months at the end of the study. Blood samples for routine lab analysis and the determination of the biomarkers were taken at baseline, after two, six and eight months. We had a dropout rate of ca. 10% (11 participants), consequently, the statistical analysis was done with the remaining 98 subjects, as shown in Appendix A. 

### 2.2. Bicycle Stress Test

Although the participants had training diaries (which were shown to be unsuitable for exact training observation [8]) we used bicycle stress tests for an exact and objective determination of the performance levels at baseline and, in particular, the performance gain after eight months of training. The bicycle stress tests were always performed with the same system (Ergometer eBike comfort, GE Medical Systems, Freiburg, Germany) in all participants with the following protocol: starting resistance of 25 watts, a 25-watts resistance increase every 2 min (according to the protocol of the Austrian Society of Cardiology, which is equal to the guidelines of the European Society of Cardiology). Blood pressure was taken every 2 min and the subjects were permanently ECG-monitored. The participants were told to cycle with 50–70 revolutions/min until exhaustion occurred. The target performance was calculated using sex, age and body surface, calculated according to DuBois formula [9]:body surface (m^2^) = 0.007184 × height (cm) ^0.725^ × weight (kg) ^0.425^(1)
A target performance of 100% represents the performance of an untrained collective. Concerning nutrition, the subjects were requested not to change their eating habits but this aspect was not controlled.

### 2.3. Lab Analysis

Blood samples for routine lab analysis, as well as the determination of sCD163 and sTWEAK, were taken at baseline, after two, six and eight months. The serum levels of soluble CD163 and TWEAK were analyzed using commercially available enzyme-linked immunosorbent assay (ELISA) kits from R&D Systems (Duoset DY1607, DY1090; R&D Systems, Minnesota, MN, USA). The preparations of plastic ware, samples, reagents and measurements were performed using the instruction manual supplied by the manufacturer. In brief, the reagents were prepared, and patient serum samples and standard protein were added to the appropriate wells of the ELISA plate (Nunc MaxiSorp flat-bottom 96 well plates, VWR International GmbH, 1150, Vienna, Austria) and were incubated for two hours. After this incubation period, the ELISA plates were washed using a Tween 20/PBS solution (Sigma Aldrich, St. Louis, MO, USA). Then, a biotin-labelled antibody was added, and the ELISA plates were incubated for another two hours. The ELISA Plates were then washed once more with Tween 20/PBS solution and a streptavidin–horseradish–peroxidase solution was added to the ELISA plate wells. For achieving a color reaction, tetramethylbenzidine for ELISA (TMB; Sigma Aldrich, St. Louis, MO, USA) was used. Optical density (OD) values were measured at 450 nm on an ELISA plate-reader (iMark Microplate Absorbance Reader, Bio-Rad Laboratories, Vienna, Austria). The coefficients of variation (CV) were: 3.4–3.8% intra-assay and 4.1–6.7% inter-assay for sCD163 and 0.1–12.8% intra-assay and 0.9–4.6% inter-assay for sTWEAK.

### 2.4. Statistical Analysis

Statistical analysis was accomplished using SPSS 26.0. Continuous and normally distributed data are described by mean ± standard deviation (std. dev.). Non-normally distributed data are described by median (25th quartile/75th quartile). To identify potential correlations between sTWEAK and sCD163 with anthropometric data or routine lab parameters at baseline, we performed a Spearman correlation using the data set of the whole population. In a second step, all of the significant correlations of the Spearman test were included in a linear regression model with the backwards method. 

To test for significant differences of sCD163 and sTWEAK levels, depending on the presence of cardiovascular risk factors, we used a Mann–Whitney U test.

To investigate the progression of the parameters of interest over the observation period, we used the Friedman test. To test for a significant difference between the levels at the beginning and the end of the observation period we used a Wilcoxon signed rank test. All tests were performed in accordance with two-sided testing and *p* values ≤ 0.05 were considered significant.

It was assumed that the initial performance level, as well as the performance gain over the observation period, would differ between the participants. For that reason, it was necessary to divide the total population into four groups depending on these two factors. Concerning the initial performance level, we chose the common cut-off at 100% to separate the group in initially unathletic and initially athletic participants. In a second step, these groups were divided dependent on their performance gain over eight months. For this separation, we chose a threshold of 3% for two reasons: first, the cut-off at 3% delivered a balanced average performance gain of about 12% in groups 2 and 4. Second, at this threshold, we observed significant changes in anthropometric and lab parameters, which are well-known to be associated with increased training. For example, groups 2 and 4 showed a significant decrease in body fat (group 2: from 31.6 to 29.7%; *p* = 0.008 and group 4: from 27.8 to 23.4%; *p* < 0.001) within the observation period. Furthermore, the HDL-cholesterol levels in group 2 increased significantly. Finally, we formed the following four groups: -Group 1: initially unathletic (initial performance < 100%), performance gain ≤ 2.9% (*n* = 9)-Group 2: initially unathletic (initial performance < 100%), performance gain > 2.9% (*n* = 32)-Group 3: initially athletic (initial performance ≥ 100%), performance gain ≤ 2.9% (*n* = 18)-Group 4: initially athletic (initial performance ≥ 100%), performance gain > 2.9% (*n* = 39)

This segmentation allows for a certain intragroup control. According to this separation, groups 2 and 4 are the “intervention” groups and groups 1 and 3 are the “controls”. Group 1 serves as a sort of control for group 2, and group 3 acts as control for group 4. 

### 2.5. Ethics Statement:

The study was carried out in adherence to the Declaration of Helsinki and its later amendments. The protocol has been approved by the Ethical Committee of the Medical University of Vienna (EC-number: 1830/2013) and informed consent was obtained from all of the participants before inclusion. 

Clinical trials registration: NCT02097199.

## 3. Results

Anthropometric and body composition data, as well as the cardiovascular risk profile, an extraction of the lab data at baseline and performance data at baseline and after eight months of training, are shown in Table 1. The most prevalent risk factors were overweight (65.9%), a positive family history for cardiovascular disease (44.9%), hypertension (32.7%) and dyslipidemia (29.6%). The high percentage of participants with T2DM (11.1%) in group 1 results from the low number of subjects in this group (*n* = 9) and, therefore, should not be overestimated. Nevertheless, there were five active smokers in group 1. To investigate the influence of the packyears on sTWEAK and sCD163, we first performed a Spearman correlation and linear regression analysis. We found no influence of the packyears on sTWEAK and sCD163 levels. In a second step we performed a Mann–Whitney U Test to compare the sTWEAK and sCD163 levels between active smokers, ex-smokers and never-smokers: never-smokers vs. ex-smokers: *p*-value sTWEAK = 0.100 and *p*-value sCD163 = 0.992; never-smokers vs. smokers: *p*-value sTWEAK = 0.347 and *p*-value sCD163 = 0.905; ex-smokers vs. smokers: *p*-value sTWEAK = 0.577 and *p*-value sCD163 = 0.880. We found no influence of the smoking status on either of the parameters. We did the same, dependent on the family anamnesis of CVD: positive vs. negative family anamnesis: *p*-value sTWEAK = 0.455 and *p*-value sCD163 = 0.983. Again, there was no significant difference.

The baseline performance of the initially unathletic groups 1 and 2 was between 87 and 89% and in the initially athletic groups 3 and 4 between 116 and 122%. However, groups 2 and 4 achieved a performance gain of ca. 12% after eight months of workout, whereas groups 1 and 3 did not achieve a significant performance gain. 

To test for correlations between sTWEAK and sCD163 with anthropometric data or routine lab parameters we performed a Spearman correlation using the data set of the whole population at baseline, as shown in Appendix A. All significant correlations of the Spearman test were included in a linear regression model with the backwards method, as shown in Table 2. For sTWEAK, the linear regression found a positive correlation with the erythrocyte count (*p* = 0.004). For sCD163, the linear regression model revealed positive correlations with the erythrocyte count (*p* = 0.005), hematocrit (*p* = 0.013), ASAT (*p* < 0.001) and lipoprotein a (*p* = 0.001).

To reveal potential differences in sCD163 and sTWEAK levels, depending on the presence of cardiovascular risk factors, we used a Mann–Whitney U test. Participants suffering hypertension had significantly higher levels of sCD163 compared to subjects without hypertension (247 (200/320) vs. 315 (243/417) ng/mL; *p* = 0.002). Subjects with a BMI < 25.0 had significantly lower sCD163 levels in comparison to participants with a BMI > 30.0 (239 (200/308) vs. 318 (244/410) ng/mL; *p* = 0.015). Females had lower sTWEAK levels compared to male participants, but the difference did not reach statistical significance (128 (97/192) vs. 156 (115/250) pg/mL; *p* = 0.064). There were neither significant differences in sCD163 nor in sTWEAK levels concerning the current smoking status and the presence of dyslipidemia.

As mentioned in the methods section, the total population has been divided into four groups, depending on their initial performance level and the performance gain over eight months. Their sTWEAK and sCD163 levels (Figure 1), as well as the results of the Friedman test and Wilcoxon signed rank test are shown in Table 3. Concerning sTWEAK, the initially unathletic group 2 with a high performance gain of > 12% showed the highest increase from 133 to 200 pg/mL (*p*-value Friedman test: 0.026; *p*-value Wilcoxon test: 0.002) whereas its “control” group 1 showed no significant change in sTWEAK levels. Similar results were obtained concerning sCD163: Participants of group 2 increased their sCD163 levels from 255 to 348 ng/mL (*p*-value Friedman test: 0.200; *p*-value Wilcoxon test: 0.035) whereas group 1 showed decreasing levels from 289 to 256 ng/mL. Concerning the initially athletic groups 3 and 4, we measured increasing sTWEAK levels in both groups, however, the increase in group 4, which showed a performance gain of > 12%, from 166 to 212 pg/mL was more pronounced and significant (*p*-value Friedman test: 0.280; *p*-value Wilcoxon test: 0.031). The sCD163 levels in group 4 increased significantly from 247 to 288 ng/mL (*p*-value Friedman test: 0.056; *p*-value Wilcoxon test: 0.025) whereas group 3 showed slightly decreasing levels.

## 4. Discussion

Regular physical workout is well known as potent strategy against atherosclerotic processes [1]. Training evolves its beneficial effects by the modulation of numerous (cardiovascular) risk factors, such as the lipid and glucose profile [10,11], and evidently ameliorates the inflammation status [12]. Since CD163 and TWEAK were associated with numerous diseases with inflammation as basic pathomechanism, the influence of training on these biomarkers and their soluble forms, sCD163 and sTWEAK, respectively, is of interest. 

In the present study we demonstrated a significant increase in sTWEAK and sCD163 due to long-term physical activity. Having a closer look at the different points of measurement, there are some interesting points that should be addressed: First, the sTWEAK levels initially increased in all groups, although the extent of the increase was more pronounced in groups 2 and 4. This might be due to a high motivation and training amount in all groups at the beginning of the study. However, in those groups who did not achieve a sufficient performance gain at the end of the study, the sTWEAK levels did not further increase, in contrast to groups 2 and 4. Second, the originally unathletic group 2 showed the most pronounced increase in sTWEAK within the first two months of training and then stabilized at a high level. Group 4 started with the highest sTWEAK levels, had a more moderate increase and finished with the highest sTWEAK amounts. Third, concerning sCD163 levels, we stated again the highest increase in group 2 but, in contrast to sTWEAK levels, this increase lagged and manifested at the end of the study. Furthermore, and this holds true for both sCD163 and sTWEAK, the initially unathletic participants of group 2 seemed to profit the most from regular physical activity as they showed the highest increase in sCD163 (> 36%) and sTWEAK (> 50%) compared to their baseline levels. Furthermore, we stated no influence of the smoking status or amount of packyears on both of the parameters. Although it is not the topic of our work, the correlation of the erythrocyte count with sTWEAK and sCD163 in the liner regression model should be at least mentioned because it might be of interest for other study groups investigating the molecular mechanisms of the CD163-mediated internalization of hemoglobin–haptoglobin complexes by CD163^+^ macrophages. 

The training-induced increase in sTWEAK levels might be of clinical relevance because several chronic diseases were associated with diminished sTWEAK amounts. In patients with chronic kidney disease, low levels of sTWEAK are associated with the presence of carotic plaques and higher cardiovascular morbidity and mortality [13]. Furthermore, sTWEAK levels are diminished in patients suffering from coronary artery disease [14], heart failure [15,16] and diabetes mellitus [17], although the mechanism leading to lower sTWEAK levels is poorly understood. A further aspect which should be referred to is the high range of sTWEAK levels in our cohort as well as in the literature. Chorianopoulos et al. [15] noted significantly lower plasma levels of sTWEAK in 364 patients suffering from heart failure compared to the controls (217 vs. 325 pg/mL). A Polish group [16] found that patients with heart failure with an ejection fraction < 35% had lower sTWEAK levels compared to comorbidities-matched controls (374 vs. 524 pg/mL). Richter et al. [18] demonstrated that low sTWEAK levels independently predict mortality in advanced non-ischemic heart failure and similar results were obtained from Urbonaviciene et al. [19] for the cardiovascular mortality of patients with peripheral artery disease. Filusch et al. found significantly lower sTWEAK levels in patients suffering from pulmonary artery hypertension compared to the controls (314 vs. 405 pg/mL), a negative association of sTWEAK with NYHA-class, pulmonary artery pressure, pulmonary vascular resistance and NT-proBNP and a positive correlation with the cardiac index and peak oxygen consumption [20]. It is well-known that physical activity prevents or retards the onset and genesis of chronic inflammatory disease, respectively (such as those mentioned above). One way that training might evolve its beneficial effects might be by increasing circulating TWEAK levels. 

CD163, which is solely expressed on the cells of the monocyte and macrophage (in particular, in the liver) lineages, is an endocytic receptor for haptoglobin–hemoglobin complexes; however, the extracellular part of CD163 can be released by ectodomain shedding by matrix metalloproteinases (MMPs) and circulate as sCD163. Thus, sCD163 levels are expected to be a marker of M2 macrophage activity [21]. Triggers for proteolytic CD163-cleavage are pro-inflammatory stimuli and oxidative stress. CD163 expression is up-regulated by glucocorticoids, IL-6, 10 and 12 and M-CSF (Macrophage colony-stimulating factor) and down-regulated by IL-4, interferon-c and TNF-α [22]. sCD163 diminishes the secretion of IL-1β and IL-6 and 8, and consequently prevents monocytes from hyperactivation [23]. Furthermore, activated M2 macrophages (which are CD163^+^) are involved in wound healing and tissue repair [24]. sCD163 levels have been shown to correlate with IL-6, which has pro- and anti-inflammatory effects [25], but not with procalcitonin and C-reactive protein [26]. Our results partly confirm these observations; however, in our cohort the sCD163 levels correlated positively with IL-6 and hsCRP only when using Spearman correlation (*p* = 0.001 and 0.042, respectively). When entering those factors in the linear regression analysis, the correlation was no longer significant. So, we found a training-induced increase in sCD163 and sTWEAK levels without increasing IL-6 or hsCRP levels. The soluble form of CD163 has been shown to reflect the amount of the receptor expressed by monocytes and macrophages, and CD163 serum levels are mostly related to M2 macrophages which have anti-inflammatory attributes and are involved in wound healing [27,28]. Similar to sTWEAK, the training-induced sCD163 increase might play a role in the sports-mediated amelioration of the inflammation status. This finding might be of interest in particular for diseases based on low-grade inflammation, such as atherosclerosis. As mentioned above, CD163-expressing macrophages reduce endogenous TWEAK expression and consequently induce the degradation of TWEAK [6]. This reaction might lead to attenuated inflammation in atherosclerosis [7].

Concerning the range of obtained sCD163 levels in our groups, as well as reported in the literature, sCD163 levels demonstrate a similar behavior to sTWEAK. Ptaszynska-Kopczynska et al. [16] reported sCD163 levels in patients with heart failure of 584 (483–665) ng/mL in comparison to their control group with 744 (570–1068 ng/mL). Llauradó et al. [29] found serum sCD163 levels of 285 (248–357) ng/mL in male patients suffering T1DM, and Moreno et al. [30] found plasma sCD163 levels of 367 (269–506) ng/mL in patients with peripheral artery disease. 

The reported high range of sTWEAK and sCD163 levels might have several reasons, such as inter alia different concentrations in serum vs. plasma, divergent methods of analysis and different ELISA kits and, of course, anthropometric, clinical or still unknown factors influencing sTWEAK and sCD163 concentrations. It seems that, at the moment, the analysis modalities are sufficient to determine values within a study group but the comparison of values derived from different study cohorts should be considered with caution. 

## 5. Conclusions

We demonstrated that regular physical activity leads to a significant increase in both sTWEAK and sCD163 serum levels. TWEAK has been shown to be diminished in patients with chronic (inflammation-based) diseases, such as coronary artery disease, heart failure, pulmonary artery hypertension, chronic kidney disease and diabetes mellitus. As is the case with many cytokines, it seems that the amount of soluble TWEAK and CD163 is essential for a healthy balance and modulation between pro- and anti-inflammatory processes, and regular physical training might use the sCD163–sTWEAK axis to unfold its beneficial effect.

## 6. Limitations

The present study has several limitations: First, although 98 participants completed the study, the number of female subjects was too low to perform a sex-specific analysis. Second, several other uncontrolled influences and circumstances (besides physical activity; e.g., nutrition, undiagnosed diseases, unreported medication, environmental aspects, etc.) might have led to an increase in sCD163 and sTWEAK levels. Third, although groups 1 and 3 evidently showed no significant change in their performance level, there were no control groups in a classical sense because, at the point of inclusion, all participants were asked to increase the extent of their physical activity. Fourth, the participants were free to choose the form of sport and the extent of this was not controlled. For that reason, bicycle stress tests were performed at the beginning and the end of the study to verify and quantify the performance gain objectively. 

## Figures and Tables

**Figure 1 jcm-09-01899-f001:**
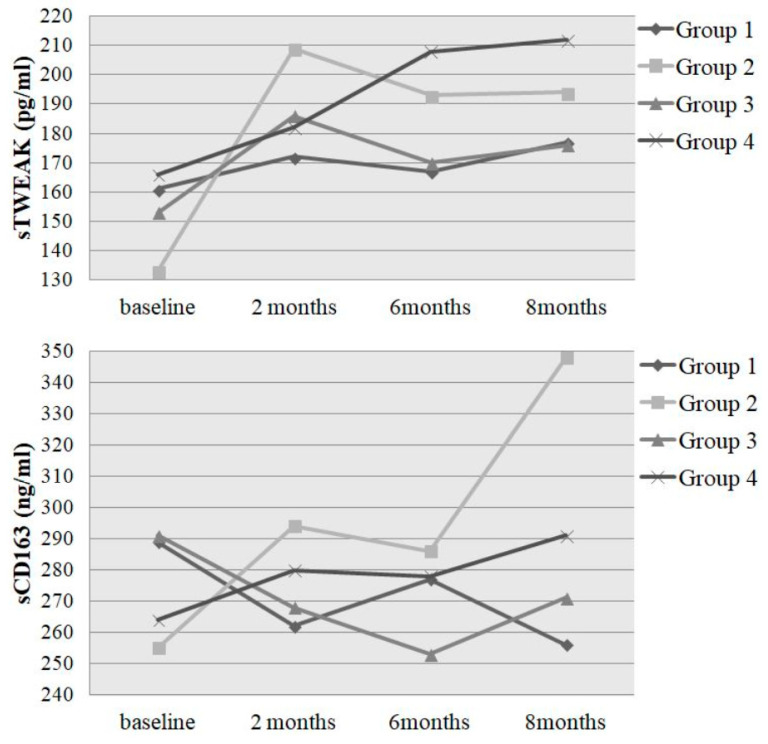
sTWEAK and sCD163 levels of the four groups over the observation time: data of the biomarkers are shown at baseline and after two, six and eight months of training. Data of the biomarkers are given as median. For greater clarity, the percentiles are not shown in the figure, but are shown in Table 3.

**Table 1 jcm-09-01899-t001:** Anthropometric and body composition data, as well as the cardiovascular risk profile, an extraction of the lab data at baseline and performance data at baseline and after eight months of training.

Parameter	Group 1 Unathletic Gain ≤ 2.9% (*n* = 9)	Group 2 Unathletic Gain > 2.9% (*n* = 32)	Group 3 Athletic Gain ≤ 2.9% (*n* = 18)	Group 4 Athletic Gain > 2.9% (*n* = 39)	Total Population
Age (years)	50.3 ± 6.1	48.6 ± 7.9	50.4 ± 6.5	49.1 ± 6.0	49.3 ± 6.7
BMI (kg/m^2^)	27.8 ± 4.2	28.5 ± 5.2	27.2 ± 3.8	26.8 ± 3.3	27.5 ± 4.2
Body fat (%)	33.9 ± 3.3	31.6 ± 6.7	26.8 ± 9.1	27.8 ± 11.8	29.4 ± 9.5
Body muscle (%)	32.4 ± 3.3	33.9 ± 4.1	34.3 ± 3.8	36.1 ± 4.0	34.7 ± 4.1
Body water (%)	48.6 ± 2.4	50.3 ± 4.9	53.8 ± 6.7	54.2 ± 5.9	52.3 ± 5.9
Performance baseline (%)	87.4 ± 9.9	88.8 ± 7.1	122.0 ± 16.8	116.0 ± 15.9	105.6 ± 19.7
Performance study end (%)	87.0 ± 9.1	101.0 ± 10.0	118.2 ± 18.0	128.2 ± 15.6	113.7 ± 20.0
Performance gain (%)	−2.7 ± 4.3	12.2 ± 7.1	−3.8 ± 4.9	12.1 ± 5.6	7.8 ± 9.1
Packyears	22.4 ± 21.4	18.9 ± 15.8	12.2 ± 9.2	16.3 ± 14.6	17.1 ± 14.9
Alcohol intake (units/week)	0.7 ± 1.0	2.8 ± 3.2	3.4 ± 4.0	3.2 ± 4.4	2.9 ± 3.8
Male sex (%)	44.4	53.1	61.1	71.8	61.2
Active smoking (%)	55.6	25.0	16.7	10.3	20.4
Positive cardiac history (%)	11.1	15.6	5.6	23.1	16.3
Diabetes mellitus (%)	11.1	3.1	5.6	0	3.1
Hypertension (%)	33.3	43.8	33.3	23.1	32.7
Dyslipidemia (%)	33.3	25.0	38.9	28.2	29.6
Overweight (%)	66.8	68.8	66.7	63.2	65.9
Positive family history (%)	66.8	43.8	50.0	38.5	44.9
Erythrocytes (T/L)	4.6 ± 0.4	4.8 ± 0.5	4.6 ± 0.4	4.7 ± 0.4	4.7 ± 0.4
Haemoglobin (g/dL)	13.3 ± 1.5	14.2 ± 1.5	13.8 ± 1.0	14.2 ± 1.2	14.0 ± 1.3
Sodium (mmol/L)	141 ± 2	141 ± 2	141 ± 2	142 ± 2	141 ± 1.7
Potassium (mmol/L)	4.2 ± 0.2	4.1 ± 0.2	4.2 ± 0.3	4.2 ± 0.2	4.2 ± 0.3
Creatinine (mg/dL)	0.8 ± 0.1	0.8 ± 0.2	0.9 ± 0.2	0.9 ± 0.2	0.9 ± 0.2
ASAT (U/L)	23 ± 4	26 ± 10	27 ± 7	24 ± 5	25 ± 7
Triglycerides (mg/dL)	154 ± 86	149 ± 100	111 ± 72	119 ± 62	131 ± 81
Cholesterol (mg/dL)	209 ± 54	200 ± 37	196 ± 29	201 ± 39	200 ± 38
HDL-cholesterol (mg/dL)	52 ± 19	56 ± 22	62 ± 12	60 ± 15	59 ± 17
LDL-cholesterol (mg/dL)	126 ± 50	117 ± 32	112 ± 29	116 ± 35	117 ± 34
HbA1c (rel.%)	5.5 ± 0.4	5.4 ± 0.8	5.5 ± 0.9	5.2 ± 0.3	5.3 ± 0.6
proBNP (pg/mL)	39 ± 27	59 ± 54	50 ± 35	32 ± 21	45 ± 39

Data are given either as mean ± std. dev. or % of population. BMI—body mass index; ASAT—aspartate aminotransferase; HDL-cholesterol—high density lipoprotein cholesterol; LDL-cholesterol—low density lipoprotein cholesterol; proBNP—pro natriuretic peptide.

**Table 2 jcm-09-01899-t002:** Linear regression model with backwards method: All significant correlations of the Spearman test, as shown in Appendix A, were included in a linear regression model with backwards method.

		Regression Coefficient B	Standard Error	β	T	Significance
sTWEAK	Constant	−474.796	233.276		−2.035	0.045
	erythrocytes	144.935	49.338	0.289	2.938	0.004
sCD163	Constant	23.408	199.495		0.177	0.907
	erythrocytes	195.298	68.091	0.454	2.868	0.005
	hematocrit	−23.627	9.282	−0.419	−2.545	0.013
	ASAT	10.772	2.334	0.428	4.615	<0.001
	lipoprotein (a)	1.018	0.287	0.314	3.547	0.001

**Table 3 jcm-09-01899-t003:** sTWEAK and sCD163 levels over the observation time: data of the biomarkers are shown at baseline and after two, six and eight months of training and their change from baseline to the end of the observation period in %. Furthermore, the results of the Friedman test and Wilcoxon signed rank test are shown. Data of the biomarkers are given as median (25th/75th percentile).

Parameter	Group 1 Unathletic Gain ≤ 2.9% (*n* = 9)	Group 2 Unathletic Gain > 2.9% (*n* = 32)	Group 3 Athletic Gain ≤ 2.9% (*n* = 18)	Group 4 Athletic Gain > 2.9% (*n* = 39)	Total Population
TWEAK baseline (pg/mL)	161 (105/213)	133 (94/216)	153 (96/209)	166 (123/214)	155 (104/213)
TWEAK 2 months	172 (132/270)	209 (111/309)	186 (145/274)	182 (133/274)	182 (135/271)
TWEAK 6 months	167 (122/215)	193 (148/255)	170 (122/267)	208 (132/288)	193 (133/253)
TWEAK 8 months	177 (123/193)	200 (152/286)	176 (140/212)	212 (151/274)	196 (148/230)
Chi^2^	0.333	9.267	6.035	3.833	15.971
*p*-value (Friedman Test)	0.954	0.026	0.110	0.280	0.001
*p*-value (Wilcoxon Test)	0.953	0.002	0.744	0.031	0.001
Change in % (baseline–end)	+9.9	+50.4	+15.0	+27.7	+26.5
CD163 baseline (ng/mL)	289 (249/354)	255 (200/403)	291 (206/351)	247 (206/325)	264 (208/332)
CD163 2 months	262 (234/323)	294 (221/437)	268 (236/391)	276 (227/353)	280 (228/356)
CD163 6 months	277 (261/370)	286 (221/419)	253 (195/390)	288 (225/358)	278 (232/383)
CD163 8 months	256 (221/331)	348 (273/396)	271 (247/359)	288 (243/394)	291 (247/384)
Chi^2^	5.000	4.644	1.800	7.560	8.032
*p*-value (Friedman Test)	0.172	0.200	0.615	0.056	0.045
*p*-value (Wilcoxon Test)	0.374	0.035	0.913	0.025	0.016
Change in % (baseline–end)	−11.4	+36.5	−6.9	+16.6	+10.2

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
