# Peer review of "Regular Training Increases sTWEAK and Its Decoy Receptor sCD163–Does Training Trigger the sTWEAK/sCD163-Axis to Induce an Anti-Inflammatory Effect?"

_jcm, 2020, doi:10.3390/jcm9061899_

Round 1

Reviewer 1 Report

The manuscript aimed to investigate the changes of sTWEAK and the receptor sCD163 after physical exercise. The sTWEAK and the receptor sCD163 levels were measured at multiple time points during the periods of study.

The results were observational. The results showed changes of sTWEAK and sCD163 after training. But there are no direct evidence for inflammatory effects. It is not convincing that these changes are related to inflammatory effects.

It is not very convincing that the increases of sTWEAK and the receptor sCD163 after physical training can be used as biomarker in clinical as the guidance for physical training based on the data shown so far in the manuscript. A few questions need to be clarified.

1, the authors need to address if a change of “155 to 196” in TWEAK or a change of “264 to 291” in CD163 can be considered clinical relevance. Papers has been published about the sTWEAK and sCD163 levels in patients. The authors should compare their value of measures with the published papers. The authors have to answer the question that if the values are still in the normal range. How these changes related to cardiovascular risks based on the published papers?

2, how did the author decide to use 2.9% gain as a cutoff for the grouping?

3, group one has more percentages of active smoking, diabetes and positive family history. It is very concerning to use it as the control group if they have such a disparity.

As discussed in the manuscript, diabetes can have lower diminished sTWEAK level according to the author (line 261-263). So the author must clarify how this disparity in groups would affect the results. The authors should also address why group 1 has higher sTWEAK than group 2 (161 vs. 133) while they have more diabetes patient?

4, the correlations of sTWEAK and sCD163 with erythrocyte counts are confusing. Are there any abnormal erythrocyte counts in these patients? Or are they just some differences within normal range? Do these correlations have clinical relevance? I am wondering what the purpose of such an analysis is.

Author Response

Comments and Suggestions for Authors

The manuscript aimed to investigate the changes of sTWEAK and the receptor sCD163 after physical exercise. The sTWEAK and the receptor sCD163 levels were measured at multiple time points during the periods of study.

The results were observational. The results showed changes of sTWEAK and sCD163 after training. But there are no direct evidence for inflammatory effects. It is not convincing that these changes are related to inflammatory effects.

It is not very convincing that the increases of sTWEAK and the receptor sCD163 after physical training can be used as biomarker in clinical as the guidance for physical training based on the data shown so far in the manuscript. A few questions need to be clarified.

Dear reviewer,

first of all we would like to thank you for your review and for giving us the opportunity to improve our manuscript according to your improvement suggestions. We know that in times of increasing workload and decreasing time resource (in particular in “Corona-times”) it is hard to find time to review a manuscript, the more we appreciate. We believe that the quality of the manuscript has clearly improved based on your comments and improvement suggestions. 

1, the authors need to address if a change of “155 to 196” in TWEAK or a change of “264 to 291” in CD163 can be considered clinical relevance. Papers has been published about the sTWEAK and sCD163 levels in patients. The authors should compare their value of measures with the published papers. The authors have to answer the question that if the values are still in the normal range. How these changes related to cardiovascular risks based on the published papers?

You are right. We have to admit that, although we cited study results dealing with the parameters in several diseases, we missed to deliver concrete values. We changed that in the manuscript.

Concerning the normal range: at the moment, we believe that the time is not yet ripe for defining standard values for the mentioned parameters. The range of published values is still simply too high. This might be caused by differences when analysing in serum vs. plasma, by using different ELISA kits, by using varying analysis methods and so on. We also added a statement concerning that topic. Thanks for drawing your attention to this point.  

Concerning the clinical relevance of the change: this is a good point. We are totally aware that statistical significance does not mean clinical relevance. However, having a look at published values of different studies (please see below) they found group differences in TWEAK-levels of about 20-30%. We found changes of 36-50% WITHIN one group (group 2) in a prospective study. So, compared to the extent of change/difference of other study populations the change in our cohorts was quite remarkable. And it is quite the same for sCD163 (we added a statement concerning the broad range of the parameters). Therefore, we believe that it might be of clinical relevance. However, due to the study design and as you wrote above, it is not possible to directly prove an influence of the changes on the inflammation status. The study should merely investigate the change of the both parameters after long-term physical activity.      

Chorianopoulos et al. (Europ. Heart Journal, 2009) stated significantly lower plasma levels of TWEAK in 364 patients suffering heart failure compared to controls (217 vs. 325 pg/ml). With other words, heart failure patients had 33% lower levels compared to controls. However, we did your analysis in serum.

A Polish group (Ptaszynska-Kopczynska et al, 2016, Cytokine) found out that patients with heart failure with an ejection fraction <35% had lower sTWEAK levels compared to comorbidities-matched controls (374 vs. 524 pg/ml). A difference of about 29%.

Urbonaviciene et al (Atherosclerosis, 2011) measured TWEAK in serum in patients with critical limb ischemia and compared these levels to those of patients with intermittent claudicatio. Again, the group with critical limb ischemia had lower levels compared to the other group (672 vs. 814, a 17% difference).

2, how did the author decide to use 2.9% gain as a cutoff for the grouping?

This is a very good point and the second reviewer had a similar question. In studies dealing with drugs, you can assume that everybody takes the exactly defined study medication as prescribed. But in sports studies, if you ask 100 participants to perform sports for weeks/months/or even years, you can be sure that a high percentage of participants will quit the study (e.g. due to loss of motivation) or do not reach a sufficiently high performance amount to really improve their performance level. So, the first problem is the “dose-problem”, which is very difficult to control. Our participants were free to choose the kind of sport and they only had to stick to the guideline “150 min. of moderate and/or 75 min. of vigorous training per week” as proposed e.g. by the European Society of Cardiology . However, it was on the dice that not all participants would reach a sufficiently high performance gain. For that reason, we performed 2 bicycle stress test (although they are much more time-consuming compared to training diaries) to determine exactly the baseline performance and the performance gain. This approach made it possible to divide the group in 4 subgroups.  

The second problem or question is: How much sport does someone has to do to 1. improve his performance objectively and 2. to have an impact on e.g. laboratory parameters of interest. The performance gain in group 2 and 4 was 12% in 8 months, which is quite much for “the average Joe”. And we received interesting results of some surrogates for sufficient (probably clinically relevant) sports-mediated performance gain at the cut off of 3% e.g.: The groups 2+4 showed a significant decrease of body fat (group 2: 31,6 to 29,7%; p=0.008 and group 4: 27,8 to 23,4%; p<0.001). Participants of group 2 could even significantly increase their HDL-cholesterol-levels, a well-known parameter which increases in sportive individuals. So, for this study, we had to find some kind of threshold and 3% seemed to be plausible based on the changes of the surrogates.   

3, group one has more percentages of active smoking, diabetes and positive family history. It is very concerning to use it as the control group if they have such a disparity.

Again a good point and we have to admit that the data presentation (in % of population) is confusing in group 1 because this group consisted of only 9 subjects. Consequently, 1 participant corresponds to 11.1% in this group. From 98 participants we only had 3 subjects suffering T2DM (1 in group 1, 1 in group 2 and 1 in group 3; the one in group 2 had insulin, the others metformin). Thus, a strong bias due to one of those participants is quite improbable.

Concerning smoking and the family anamnesis, it is a bit different. We had 5 active smokers in group 1 and you are totally right that this high prevalence might have had an influence on the data. For that reason, we re-analysed the data set: there was no correlation (Spearman and linear regression) of the packyears with baseline sCD163 or sTWEAK-levels. In a second step, we performed a Mann-Whitney U Test and compared sCD163 and sTWEAK-levels dependent on the present smoking status:

Never-smokers vs. ex-smokers: p-value sTWEAK=0,100; p-value sCD163=0,992

Never-smokers vs. smokers: p-value sTWEAK=0,347; p-value sCD163=0,905

Ex-smokers vs. Smokers: p-value sTWEAK=0,577; p-value sCD163=0,880

Pos. vs. neg. family anamnesis: p-value sTWEAK=0,455, p-value sCD163=0,983 

Thus, in our cohort we did not find an influence of packyears, present smoking status or a positive family anamnesis on sTWEAK or CD163-levels and a bias due to the smoking status in our cohort is quite improbable. However, the influence of those parameters has to be further investigates in further studies with large cohorts, suitable for highly diagnostic multivariate analysis.        

As discussed in the manuscript, diabetes can have lower diminished sTWEAK level according to the author (line 261-263). So the author must clarify how this disparity in groups would affect the results. The authors should also address why group 1 has higher sTWEAK than group 2 (161 vs. 133) while they have more diabetes patient?

Please see the statement above: we only had 1 T2DM-patient in group 1, 1 in group 2 and 1 in group 3. The number of T2DM war very low and “distributed” on three groups, therefore, a relevant bias is improbable.

4, the correlations of sTWEAK and sCD163 with erythrocyte counts are confusing. Are there any abnormal erythrocyte counts in these patients? Or are they just some differences within normal range? Do these correlations have clinical relevance? I am wondering what the purpose of such an analysis is.

Again an interesting point. The erythrocyte count in our population was normal. Concerning the purpose of the analysis: as TWAEK and CD163 are still researched intensively (in particular their clinical aspects) we considered it necessary to reveal every positive or negative correlation of these factors with anthropometric or lab-data. As shown in Suppl. Table 1, we found several correlations but only a few were still significant in the linear regression model. A correlation of TWEAK with the white blood cell count and platelets has already been shown (e.g. doi: 10.1016/j.cyto.2016.02.005, doi: 10.3109/00365513.2011.629678, doi: 10.1111/j.1744-9987.2011 but the correlation of TWEAK with erythrocytes is a new finding. However, due to the concept and the topic of the study and the available data, we are not able to sufficiently interpret or evaluate this finding. Of course it would be interesting to further analyse regulators of erythropoiesis (e.g. Vitamin B12, folic acid, erythropoietin…) in connection to TWEAK but those factors had not been analysed. Nevertheless, we think it is necessary to at least demonstrate that this correlation existed in our cohort. Concerning the correlation of CD163 with haematocrit and erythrocytes our results may be of interest for groups investigating the molecular mechanisms of CD163 mediated internalisation of haemoglobin-haptoglobin complexes by macrophages. E.g. recently it has been shown that CD163+-macrophages are able to recycle free haemoglobin and to take up senescent red blood cells (Hu et al. 2019; nutrients).

Thanks again for your review. We appreciate your effort. If you have further questions or improvement suggestions do not hesitate to contact us.    

Reviewer 2 Report

I reviewed this manuscript with interest. The authors show that a long period of repeated physical activity reduces the inflammatory status of the participants of their study. This was done by determination of the sTWEAK and sCD163 levels during the study. The study seems to be performed well and the authors are aware that the physical activity that was applied was not under sufficient control. Therefore, they performed a bicycle test only at the beginning (baseline) and after 8 months (maximal effect?). At different points during the study they collected blood samples, which were analysed and the results were reported in the study together with the different time points of sTWEAK and sCD163 levels. In their study the authors observed an increase of sTWEAK and sCD163 with physical activity especially in the groups which responded most to 8 months of physical activity. The manuscript displays several shortcomings, which I would like to explain now.

  1. The bicycle test: where does the number of 2.9 % gain increase come from? How is the gain determined? What is the target performance? Which subjects are considered for the 100% (untrained collective)? Does this mean that the bicycle test delivered the same results at baseline and after 8 months? Did you find any differences between subjects claiming to have followed the moderate and vigorous physical activity program?
  2. Is the fact that the four groups start at different baselines a problem? Bias? Is the inflammatory status of all participants comparable. I assume it is not. Can you estimate this (WBC's?). Why did you choose to divide the population in 4 groups depending on the results of the bicycle test? Did you expect that untrained individuals (performance test < 100%) would show lower sTWEAK and sCD163 levels than "trained" individuals (performance test > 100%), which is not true. How does this fit within the idea that physical activity increases sTWEAK and sCD163 levels. 
  3. In the abstract-results no data are shown for the time-dependency of the sTWEAK and sCD163 levels.
  4. You measured heart rate and BP. How where these parameters changed in the four groups of "patients"? I assume this contributes somehow to the performance gain?
  5. Do you have an idea how the 8 months sTWEAK and sCD163 values of the groups responding most to physical activity correspond with both serum values in a healthy population?
  6. In the discussion, I expected some lines on the scavenger receptor activity of CD163 for TWEAK. How are they related to each other, knowing that the values may change especially after 2 months of physical activity for sTWEAK. Time-dependency for sCD163 (decreasing for groups 1 and 3, increasing for groups 2 and 4) is different from time-dependency of sTWEAK (always increasing).
  7. Participants with hypertension had higher sCD163 levels.This is interesting but not discussed in the manuscript. What did physical activity do here? Why are sCD163 levels higher (link with inflammation?) in hypertension, compared with normotension and how to see this in relation with the increased values in positive gain participants of the present study? 

Minor remarks:

  1. In the study population I would start with inclusion criteria and later tell which criteria were excluded.
  2. line 171 contains . and :
  3. line 189, the reason for including AST
  4. table 1: why not put active smoker and packyears together?
  5. line 236: is well known as potent means (reword)
  6. line 282: our results

Author Response

Comments and Suggestions for Authors

I reviewed this manuscript with interest. The authors show that a long period of repeated physical activity reduces the inflammatory status of the participants of their study. This was done by determination of the sTWEAK and sCD163 levels during the study. The study seems to be performed well and the authors are aware that the physical activity that was applied was not under sufficient control. Therefore, they performed a bicycle test only at the beginning (baseline) and after 8 months (maximal effect?). At different points during the study they collected blood samples, which were analysed and the results were reported in the study together with the different time points of sTWEAK and sCD163 levels. In their study the authors observed an increase of sTWEAK and sCD163 with physical activity especially in the groups which responded most to 8 months of physical activity. The manuscript displays several shortcomings, which I would like to explain now.

Dear reviewer,

first of all we would like to thank you for your review and for giving us the opportunity to improve our manuscript according to your improvement suggestions. We know that in times of increasing workload and decreasing time resource (in particular in “Corona-times”) it is hard to find time to review a manuscript, the more we appreciate. We believe that the quality of the manuscript has clearly improved based on your comments and improvement suggestions. 

  1. The bicycle test: where does the number of 2.9 % gain increase come from? How is the gain determined? What is the target performance? Which subjects are considered for the 100% (untrained collective)? Does this mean that the bicycle test delivered the same results at baseline and after 8 months? Did you find any differences between subjects claiming to have followed the moderate and vigorous physical activity program?

This is a very good point and the second reviewer had a similar question. In studies dealing with drugs, you can assume that everybody takes the exactly defined study medication as prescribed. But in sports studies, if you ask 100 participants to perform sports for weeks/months/or even years, you can be sure that a high percentage of participants will quit the study (e.g. due to loss of motivation) or do not reach a sufficiently high performance amount to really improve their performance level. So, the first problem is the “dose-problem”, which is very difficult to control. Our participants were free to choose the kind of sport and they only had to stick to the guideline “150 min. of moderate and/or 75 min. of vigorous training per week” as proposed e.g. by the European Society of Cardiology . However, it was on the dice that not all participants would reach a sufficiently high performance gain. For that reason, we performed 2 bicycle stress test (although they are much more time-consuming compared to training diaries) to determine exactly the baseline performance and the performance gain. This approach made it possible to divide the group in 4 subgroups. 

The second problem or question is: How much sport does someone has to do to 1. improve his performance objectively and 2. to have an impact on e.g. laboratory parameters of interest. The performance gain in group 2 and 4 was 12% in 8 months, which is quite much for “the average Joe”.

To answer your first question: it is quite hard to define a threshold for a “significant” performance gain, thus, we oriented on some surrogates, which are well-known to change due to log-term physical activity. At the cut off of 3% the “intervention” groups 2+4 showed a significant decrease of body fat (group 2: 31,6 to 29,7%; p=0.008 and group 4: 27,8 to 23,4%; p<0.001). Participants of group 2 could even significantly increase their HDL-cholesterol-levels, a well-known parameter which increases in sportive individuals. So, for this study, we had to find some kind of threshold and 3% seemed to be plausible based on the changes of the surrogates.

How is the gain determined? The determination was calculated as follows: performance after 8 months of training minus baseline performance. The “intervention groups showed about 12% performance gain within 8 months of training.

What is the target performance? Basically, we have to distinguish 3 formulas which are important in our study: The formula for the calculation of the body surface (DuBois), the formula to calculate the target Watt-peak and the formula to calculate a training pulse (Karvonen). The target performance (=the amount of Watts that an individual should be able to reach) mainly depends on the body surface area (BSA). The calculation of the BSA has a long history of more than 100 years and to our knowledge DuBois was the first to propose a formula in 1916. So, the first important part calculating the target performance is the BSA. The other parameters are sex and age (apart from the modulating factors). Based on these parameters, you can calculate a target Watt-amount that should be reached by the subject. Either you have a chart to look it up or the ergometer calculates the target Watt-amount automatically if you enter age, sex, height and weight. Of course you can calculate it manually (e.g. for male participants: 6,773 + 136,141 * BSA * age). The third formula is the Karvonen-fomula [training pulse= baseline heart rate + (max. heart rate – baseline heart rate) * training factor]. Based on this formula you can calculate a training pulse.

Which subjects are considered for the 100% (untrained collective)? Thanks for that question. It is a general agreement in sport medicine to classify subjects with a performance 100% as the untrained “average Joe”. It makes a great difference whether you initially have a “sports-affine” participant or someone who has not done any sports before participating in the study. When developing the study protocol, it was clear that we have to make 1. a separation of already well-trained participants and not trained subject. This is the reason why we divided the subjects initially in unathletic and athletic ones. 2. (and this is in particular important for the study question) it was necessary to separate in each group those subjects who did not sufficiently exercise (somehow a kind of control groups) from those who DID perform sufficiently and increased their performance level objectively (the “intervention” group). If we had not taken those separations, we would have “compared apples to oranges”.  

Does this mean that the bicycle test delivered the same results at baseline and after 8 months? Well, to obtain comparable results all circumstances (the bicycle stress test protocol as well as the ergometer itself and the exercise protocol…) were always the same from the beginning to the end in every participant. The target performance was calculated individually. And indeed we had participants whose performance was nearly the same at the beginning and the end of the study. However, this was expectable and of advantage for us because we even used this fact to form “control groups”.

Did you find any differences between subjects claiming to have followed the moderate and vigorous physical activity program? This point is of particular interest. Of course many sports studies rely on training diaries because they are easy to handle and cheap but they are also very vague and subjective. Despite the high effort, we decided to perform bicycle stress test at the beginning and the end of the study to define the baseline performance level and to prove and quantify the performance gain exactly. The participants had guidelines how to classify the sports modality: e.g. quick walking, Nordic walking, slow cycling/swimming, (inline) skating, hiking were defined as moderate intensity training and soccer, tennis, running/jogging, quick cycling/swimming, paddling were defined as high intensity training. And of course we also had training diaries. To return to you question: The entries of the participants in the training diaries BARELY correlated with the objectively measured performance gain in our study. The bicycle stress test is the gold standard in exercise testing, in particular if you want to use these data for scientific purposes. To our mind, training diaries may be used in addition but we would never rely on their results in study matters.      

  1. Is the fact that the four groups start at different baselines a problem? Bias? Is the inflammatory status of all participants comparable. I assume it is not. Can you estimate this (WBC's?). Why did you choose to divide the population in 4 groups depending on the results of the bicycle test? Did you expect that untrained individuals (performance test < 100%) would show lower sTWEAK and sCD163 levels than "trained" individuals (performance test > 100%), which is not true. How does this fit within the idea that physical activity increases sTWEAK and sCD163 levels.

A good question and again you address a point that has partly been addressed by reviewer 2 too. In literature, the levels of TWEAK and CD163 have a high range (probably due to different modalities of analysis, different media serum/plasma, different ELISA-kits…) and we added a corresponding statement in the manuscript. Concerning TWEAK in our cohort: the groups 1+3+4 started at very similar levels, only group 2 started with lower levels. However, despite the fact that group 2 started with lower levels they reached much higher levels compared to group 1+3 (which were the groups without significant performance gain). Concerning CD163, you are right that group 1+3 started with higher baseline levels compared to group 2+4 but nevertheless, those who had a performance gain >3% could significantly increase their levels but those without a gain did not. In particular in group 2 the CD163-increase was impressive.   

Concerning the WBC at the beginning: group 1: 7,5 G/l; group 2: 6,7 G/l; group 3: 6,5 G/l; group 4: 6,2 G/l. So, concerning the WBC the groups were within the normal range. Furthermore, we performed a Mann-Whitney-U test for the WBC of group 1 vs. 3 (p=0,164) and of group 2 vs. 4 (0.152). Again there was no significant difference at the beginning in WBC.

Concerning the division into 4 groups: we hope we could explain our approach in the statements above. In short, we had to determine 1. Who was already initially sportive and 2. who could reach a significant performance gain. Based on these 2 circumstances, we formed 4 groups. Group 2 (“controlled” by group 1) and group 4 (“controlled” by group 3). Otherwise, a proper comparison would not have been possible.   

Concerning the last point and concerning TWEAK: Actually we had no expectation about the baseline levels because most data available (mostly in patients with heart failure, peripheral artery disease, pulmonary hypertension, dialysis…so, really ill patients) showed a high range as mentioned before. Although the subjects in our cohort had several cardiovascular risk factors, they were quite “healthy” compared to above mentioned patients. Thus, we did not expect a great difference. Concerning CD163 we found median CD163-levels of about 255 to 290. Of course this is a certain difference but compared to the ranges of other reported cohorts it is quite low. And concerning the interpretation: CD163 might be a marker involved in inflammation but actually, “controlled” inflammation is absolutely essential. It has been shown that CD163 prevents monocytes from hyperactivation and activated M2 macrophages (which are CD163 positive) are involved in wound healing. Thus, we caution against the assumption that CD163 is just a proinflammatory cytokine. In fact, we believe that CD163 is an important marker for a healthy balance and modulation of pro and anti-inflammatory processes.

  1. In the abstract-results no data are shown for the time-dependency of the sTWEAK and sCD163 levels.

Yes, this was a weak point of the manuscript and we added the levels. Thanks for drawing our attention to that.

  1. You measured heart rate and BP. How where these parameters changed in the four groups of "patients"? I assume this contributes somehow to the performance gain?

Yes, all bicycle stress tests were ECG-monitored and BP was measured every 2 minutes. However, HR or BP do not influence the performance gain, meaning that we do not abort the test just because of a HR of 190 bpm or a BP >220 mmHg. All test were done until exhaustion occured. And concerning a change over time you are right. Group 2 showed lower diastolic BP values at the second ergometrie (probably also due to sufficient training). In the other groups we did not find any significant changes.  

  1. Do you have an idea how the 8 months sTWEAK and sCD163 values of the groups responding most to physical activity correspond with both serum values in a healthy population?

As mentioned above, both parameters show high ranges. In general, our cohort had lower levels compared to others but our participants were “less ill” compared to others. Data of really healthy individuals are quite rare but some are available: E.g. Llaurado et al (2012) found serum sCD163-levels of 285 (248-357) ng/ml in male patients suffering T1DM and compared them at least to healthy sex-matched subjects < who had 225 (193.3–296.5) ng/ml. But the same group found much higher TWEAK-levels (also in the healthy control group) compared to our cohort. That ist he reason why we believe that an „intra-study“ comparison of the levels should be interpreted with caution.  

  1. In the discussion, I expected some lines on the scavenger receptor activity of CD163 for TWEAK. How are they related to each other, knowing that the values may change especially after 2 months of physical activity for sTWEAK. Time-dependency for sCD163 (decreasing for groups 1 and 3, increasing for groups 2 and 4) is different from time-dependency of sTWEAK (always increasing).

Thanks for drawing our attention to that, indeed this point went short and we added some information.

  1. Participants with hypertension had higher sCD163 levels.This is interesting but not discussed in the manuscript. What did physical activity do here? Why are sCD163 levels higher (link with inflammation?) in hypertension, compared with normotension and how to see this in relation with the increased values in positive gain participants of the present study? 

This is of course an interesting point. However, we advise again overinterpreting this correlation. For this analysis we basically used anamnestic data and hypertension medication respectively. And nearly all of our participants with hypertension had a well-controlled blood pressure and in case it was not well-controlled, we initially prescribed corresponding medication. So, this correlation is really only based on the diagnosis „hypertension“ and not on BP-values. Nevertheless, this would be a good topic for future studies but in this case we ad to detect numerous BR-values (preferably by 24h-BP-measurement) and of course perform a multivariat analysis including all hypertension medication.  

Minor remarks:

  1. In the study population I would start with inclusion criteria and later tell which criteria were excluded.

Yes, thanks for the hint. We changes that in the manuscript. Sounds better now.

  1. line 171 contains . and :

We found the mistake. Thanks

  1. line 189, the reason for including AST

Again an interesting point and the other reviewer asked a similar question concerning erythrocytes. Table 1 should just give an overview of the cohort so we included some values of the blood count, renal function, liver function…. Concerning the purpose of the analysis: as TWAEK and CD163 are still researched intensively (in particular their clinical aspects) we considered it necessary to reveal every positive or negative correlation of these factors with anthropometric or lab-data. As shown in Suppl. Table 1, we found several correlations but only a few were still significant in the linear regression model. AST was one of them. This might be of little relevance for our research topic but maybe it is of interest for other groups, for that reason we at least mentioned it.

  1. table 1: why not put active smoker and packyears together?

Good point. We re-analysed the data set: there was no correlation (Spearman and linear regression) of the packyears with baseline sCD163 or sTWEAK-levels. In a second step, we performed a Mann-Whitney U Test and compared sCD163 and sTWEAK-levels dependent on the present smoking status:

Never-smokers vs. ex-smokers: p-value sTWEAK=0,100; p-value sCD163=0,992

Never-smokers vs. smokers: p-value sTWEAK=0,347; p-value sCD163=0,905

Ex-smokers vs. Smokers: p-value sTWEAK=0,577; p-value sCD163=0,880

Pos. vs. neg. family anamnesis: p-value sTWEAK=0,455, p-value sCD163=0,983 

Finally, we did not find a significant influence of the smoking status on the both parameters. However, we had active smokers smoking 3 cigarettes/day but also some smoking 20/day. We thought that distinguishing between smoking status and packyears would deliver more detailed data.

  1. line 236: is well known as potent means (reword)

Yes, we reworded this passus.

  1. line 282: our results

Thanks, result was changed into results.

Thanks again for your review. We appreciate your effort. If you have further questions or improvement suggestions do not hesitate to contact us.   

Round 2

Reviewer 1 Report

The authors have made very good efforts addressing the questions in the Author’s notes. They are clear and powerful. But I am wondering why most of the good comments from the Author’s notes were not reflected in the manuscript.

1, about the comparison of the TWEAK and CD163,

The authors have made clear points both in the comments from the Author’s notes and the manuscript. It is a good job.

2, about the cutting point of 2.9% gain,

The authors explained well about this point in the comments from the Author’s notes. But this should also be included in the manuscript for the readers.

It would be better to see those information in the method part of the manuscript.

3, about the question of group one having more percentage of active smoking, diabetes and positive family history,

The information provided from the Author’s notes are detailed and clear. Again, I am wondering why the authors didn’t put them in the manuscript. Those information and explanations from the Author’s notes are very important for the readers to understand the data.

4, about correlations of sTWEAK and sCD163 with erythrocyte counts,

The authors explained their opinions in the Author’s notes. People may have different thoughts about these data. It is OK to have the data included in the manuscript. But the authors should include their explanations about the correlation from the author’s notes into the discussion part of the manuscript. Again, the authors need to put their opinions in the Author’s notes into the manuscript for the readers.

After the clarification from the author's notes, I am convinced the study is good. I hope the authors can incorporate the detailed information from the comments of the Author’s notes to the manuscript for the readers. I will be glad to see the further edited manuscript.

Author Response

The authors have made very good efforts addressing the questions in the Author’s notes. They are clear and powerful. But I am wondering why most of the good comments from the Author’s notes were not reflected in the manuscript.

Again, the authors express their gratitude for the constructive improvement suggestions. We have to admit that we were very focused on the point-by-point responses that we sometime forgot to include the main message in the manuscript. We made up for it.  

1, about the comparison of the TWEAK and CD163,

The authors have made clear points both in the comments from the Author’s notes and the manuscript. It is a good job.

2, about the cutting point of 2.9% gain,

The authors explained well about this point in the comments from the Author’s notes. But this should also be included in the manuscript for the readers.

It would be better to see those information in the method part of the manuscript.

Yes. We added the following explanation in the results section: It was to assume that the initial performance level as well as the performance gain over the observation period would differ between the participants. For that reason, it was necessary to divide the total population into 4 groups depending on these two factors. Concerning the initial performance level we chose the common cut-off at 100% to separate the group in initially unathletic and initially athletic participants. In a second step, these groups were divided dependent on their performance gain over 8 months. For this separation, we chose a threshold of 3% for 2 reasons: first, the cut-off at 3% delivered a balanced average performance gain of about 12% in group 2 and 4. Second, at this threshold, we observed significant changes in anthropometric and lab parameters which are well-known to be associated with increased training: e.g. the groups 2 and 4 showed a significant decrease of body fat (group 2: from 31.6 to 29.7%; p=0.008 and group 4: from 27.8 to 23.4%; p<0.001) within the observation period. Furthermore, the HDL-cholesterol-levels in group 2 increased significantly. Finally, we received the following 4 groups:

3, about the question of group one having more percentage of active smoking, diabetes and positive family history,

The information provided from the Author’s notes are detailed and clear. Again, I am wondering why the authors didn’t put them in the manuscript. Those information and explanations from the Author’s notes are very important for the readers to understand the data.

Yes. We added the following in the results section (and a shorter version in the discussion): The high percentage of participants with T2DM (11.1%) in group 1 results from the low number of subjects in this group (n=9) and should not be overestimated. Nevertheless, there were 5 active smokers in group 1. To investigate the influence of the packyears on sTWEAK and sCD163 we first performed a Spearman correlation and linear regression analysis. We found no influence of the packyears on sTWEAK and sCD163-levels. In a second step we performed a Mann-Whitney U Test to compare the sTWEAK and sCD163-levels between active smokers, ex-smokers and never-smokers: never-smokers vs. ex-smokers: p-value sTWEAK=0.100 and p-value sCD163=0.992; never-smokers vs. smokers: p-value sTWEAK=0.347 and p-value sCD163=0.905; ex-smokers vs. smokers: p-value sTWEAK=0.577 and p-value sCD163=0.880. We found no influence of the smoking-status on the both parameters. We did the same dependent on the family anamnesis of CVD: positive vs. negative family anamnesis: p-value sTWEAK=0.455 and p-value sCD163=0.983. Again, there was no significant difference.

4, about correlations of sTWEAK and sCD163 with erythrocyte counts,

The authors explained their opinions in the Author’s notes. People may have different thoughts about these data. It is OK to have the data included in the manuscript. But the authors should include their explanations about the correlation from the author’s notes into the discussion part of the manuscript. Again, the authors need to put their opinions in the Author’s notes into the manuscript for the readers.

Again we added a statement in the discussion section.

After the clarification from the author's notes, I am convinced the study is good. I hope the authors can incorporate the detailed information from the comments of the Author’s notes to the manuscript for the readers. I will be glad to see the further edited manuscript.

Reviewer 2 Report

The authors carefully responded to my comments and suggestions, for which I would like to thank them. Minor remarks:

line 55 das = was

Abstract: inclusion of data is OK, but ad the physical activity time period in the sentence (8 months).

Author Response

The authors carefully responded to my comments and suggestions, for which I would like to thank them. Minor remarks:

line 55 das = was

DONE

Abstract: inclusion of data is OK, but ad the physical activity time period in the sentence (8 months).

DONE

Again, the authors express their gratitude for the constructive improvement suggestions.
